# Tomographic imaging of the photonic environment of plasmonic nanoparticles

Anton Hörl[1], Georg Haberfehlner[2,3], Andreas Trügler [1], Franz-Philipp Schmidt[1,2,3], Ulrich Hohenester[1] & Gerald Kothleitner[2,3]

The photonic local density of states (LDOS) governs the enhancement of light–matter interaction at the nanoscale, but despite its importance for nanophotonics and plasmonics experimental local density of states imaging remains extremely challenging. Here we introduce a tomography scheme based on electron microscopy that allows retrieval of the three-dimensional local density of states of plasmonic nanoparticles with nanometre spatial and sub-eV energy resolution. From conventional electron tomography experiments we obtain the three-dimensional morphology of the nanostructure, and use this information to compute an expansion basis for the photonic environment. The expansion coefficients are obtained through solution of an inverse problem using as input electron-energy loss spectroscopy images. We demonstrate the applicability of our scheme for silver nanocuboids and coupled nanodisks, and resolve local density of states enhancements with extreme sub-wavelength dimensions in hot spots located at roughness features or in gaps of coupled nanoparticles.

---

[1] Institute of Physics, University of Graz, Universitätsplatz 5, 8010 Graz, Austria. [2] Graz Centre for Electron Microscopy, Steyrergasse 17, 8010 Graz, Austria. [3] Institute for Electron Microscopy and Nanoanalysis, Graz University of Technology, Steyrergasse 17, 8010 Graz, Austria. Anton Hörl and Georg Haberfehlner contributed equally to this work.   Correspondence and requests for materials should be addressed to U.H. (email: ulrich.hohenester@uni-graz.at)

Plasmonics allows confinement of light at the nanoscale. This is achieved by binding light to coherent electron charge oscillations at the boundary of metallic nanoparticles, so-called particle plasmons, which come together with strong and tightly focused electromagnetic nearfields[1–3]. While light confinement to extreme subwavelength dimensions holds promise for many applications, it hinders direct observation of the plasmonic nearfields by optical means because of the diffraction limit of light. An alternative measurement scheme is provided by electron energy-loss spectroscopy (EELS) in a scanning transmission electron microscope (STEM), where a focused beam of swift electrons passes by or through metallic nanoparticles and the electrons can lose a small fraction of their kinetic energy through excitation of particle plasmons. By raster scanning the electron beam over the specimen and spectrally analyzing the electron energy-loss, one can map the plasmonic nearfields with nanometre spatial and sub-eV energy resolution. This has been used intensively to investigate particle plasmons in various nanostructures, and has for instance allowed direct observation of coupling and quantum effects or optically dark modes[4–9].

Although the electron beam is a local probe, interpretation of the EELS signal is not straightforward. First of all the electron energy-loss is determined by the integral effect of plasmonic nearfields along the beam path, thus masking information along the beam propagation direction. Second there is no generally valid simple link between a local physical quantity and the EELS signal. Though it was suggested that EELS might directly probe the photonic local density of states (LDOS)[10], a quantity of immense importance in nanophotonics, it was later shown that this holds only for special cases[11].

Tomographic reconstruction can in principle retrieve the three-dimensional (3D) distribution of an unknown physical quantity from a series of projections with different viewing angles, under the assumption that probe–sample interactions are local in space and fulfill the projection requirement[12]. Generally in the case of EELS the energy loss of electrons to plasmons is a non-local process, where the swift electron excites a plasmon oscillation and subsequently performs work against the induced plasmon field[13]. However, under certain restrictive assumptions, such as the quasistatic approximation, applicable for nanoparticles much smaller than the wavelength of light, and a plasmonic response governed by a single resonance, one can formulate a conventional tomography scheme[14]. For the quasistatic case, the EELS signal can be interpreted as a probe for the electrostatic potential along the beam trajectory and a three-dimensional reconstruction of the potential has been demonstrated for a silver nanocube[15]. In the same context it is also possible to trace the electrostatic potential back to surface charges[14], which has also been demonstrated experimentally[16]. Unfortunately, these approaches come with several shortcomings (e.g., the schemes of refs. [14, 15] require a plasmonic response governed by a single mode and electron trajectories that do not penetrate the nanoparticle) and, more importantly, fail for larger nanoparticles beyond the quasistatic approximation[14, 17]. In addition to EELS, cathodoluminescence (CL) is another technique based on electron microscopy that provides complementary information about the photonic environment[18–20]. A comparative study between EELS and CL has been presented in ref. [21], and a tomography study using CL is given in ref. [22].

Here we demonstrate tomographic reconstruction of plasmonic nearfields through the solution of an inverse problem, following a recent proposal[17], and prove the applicability of this scheme using experimental EELS maps obtained for various nanoparticle geometries.

## Results

**Tomography scheme.** The basic principle underlying our approach is shown in Fig. 1 and can be roughly divided into three steps. In the first step, we perform a tomographic experiment measuring EELS spectrum images (SIs) and high-angle annular dark-field (HAADF) images for different tilt angles. From the HAADF data, we reconstruct the particle geometry, using a conventional tomography scheme[23, 24], and use the reconstructed particle geometry to compute for each loss energy the complex-valued plasmonic eigenmodes $E_k(\mathbf{r})$[17, 25]. A truncated set of these modes serves as a generic basis for the decomposition of the Green tensor[25]

$$G(\mathbf{r}, \mathbf{r}') \approx \sum_{k=1}^{n} C_k E_k(\mathbf{r}) \otimes E_k(\mathbf{r}'), \qquad (1)$$

which, for given expansion coefficients $C_k$, provides a complete characterization of the photonic environment of the plasmonic nanoparticle[17, 26]. In a second step, we determine the coefficients $C_k$ using a minimization procedure. For some initial guess of $C_k$, we compute the reprojected loss probabilities $P_{rep}(\mathbf{R}, \theta)$ for different beam positions $\mathbf{R}$ and tilt angles $\theta$, using the standard EELS equations[13], and then update $C_k$ such that

$$\min_{C_k} \left[ \|C_k\|_{L_1} + \frac{1}{2\mu} \|P_{exp} - P_{rep}\|_{L_2}^2 \right] \qquad (2)$$

becomes minimized, where $P_{exp}$ denotes the experimental EELS maps. In short, the second expression is minimal when the reprojected maps resemble the experimental ones as closely as possible, whereas the first expression favors a Green function decomposition with as few nonzero expansion coefficients as possible. Such a bias is in the spirit of compressed sensing optimizations[27, 28] that have proven successful in a large number of applications including different problems in electron tomography[15, 29–31]. $\mu$ is a parameter that weights between these two minimization objectives (for details see Methods). Once we have determined the best expansion coefficients $C_k$, we have completely characterized the photonic environment of the plasmonic nanoparticle and can compute in a third step all quantities of interest, such as for instance the photonic LDOS.

It is worth emphasizing that the plasmonic eigenmodes $E_k(\mathbf{r})$ are solutions of the full Maxwell equations, thus our approach is suited for larger nanoparticles. As these modes provide a complete basis, similar to the Mie solutions for a spherical particle[2], in principle any photonic environment can be expressed in terms of Eq. (1) provided that the cutoff parameter $n$ is chosen sufficiently large. However, to render Eq. (1) suitable for a compressed sensing optimization it is necessary that only a few coefficients $C_k$ differ significantly from zero, which can be achieved by using an expansion basis $E_k(\mathbf{r})$ well adapted to the problem (as we do by computing the modes for the tomographically reconstructed particle geometry).

**Experiments.** To demonstrate the applicability of our tomographic reconstruction scheme, we perform experiments on two types of plasmonic nanostructures created by electron-beam lithography on a thin $Si_3N_4$ membrane: first a silver nanocuboid (dimensions $300 \times 140 \times 30$ nm³), and second two coupled silver nanodisks (diameter 180 nm, thickness 30 nm, gap 25 nm). While the nanocuboid serves as a proof of principle, investigations of the coupled disks allow direct visualization of coupling effects.

In the microscope, we acquire HAADF STEM images and EELS SIs at different tilt angles. The 3D morphology of the nanostructure is reconstructed using a total-variation minimization algorithm[24, 29, 31]. To extract plasmon resonance maps,

spectra are pre-treated with Richardson–Lucy deconvolution (Supplementary Figs. 1 and 2) and are integrated over a range of 0.19 eV around each resonance energy for each pixel of the SIs. The 3D reconstruction of the nanoparticle shape is used to simulate energy-loss spectra and resonance maps for each tilt angle[24, 32], which in turn can be compared to experimental data.

For the nanocuboid sample, we identify two prominent resonance peaks in the loss spectra of Fig. 2a, which can be attributed to the dipolar and quadrupolar surface plasmon modes. The case of the coupled nanodisks is more complicated and several resonance peaks are visible in Fig. 2b; the lowest energy mode appears strongly localized at a protrusion attached to one of the nanodisks, the other ones are attributed to coupled edge modes of the disks[9].

**LDOS reconstruction.** Using the reconstructed morphology and angle-dependent plasmon resonance maps, we employ the scheme proposed above to determine the "best" coefficients $C_k$ and to reconstruct the dyadic Green tensor **G** of Eq. (1) (see Supplementary Figs. 3 and 4 for a comparison of experimental and reprojected EELS maps). From **G** we can compute any electrodynamical quantity of interest, such as the projected photonic LDOS[26]

$$\rho_{\mathbf{n}}(\mathbf{r}) = \frac{6\omega}{\pi c^2} \mathrm{Im}\{\mathbf{n} \cdot \mathbf{G}(\mathbf{r},\mathbf{r}) \cdot \mathbf{n}\}, \qquad (3)$$

where $\omega$ is the angular frequency and $c$ the speed of light. Similar to solid-state physics, where the LDOS provides the number of states accessible to electrons per unit energy and volume, the photonic LDOS provides the density of photonic states and

describes how strongly a quantum emitter (e.g., fluorescent molecule or quantum dot) with dipole moment $d$ oriented along direction **n** couples to the photonic environment[10]. The decay rate of a dipole located at position **r** can be expressed as $\Gamma_{\mathrm{dip}} = (2\omega d^2)/(3\hbar\varepsilon_0)\rho_{\mathbf{n}}(\mathbf{r})$, with $\hbar$ being the reduced Planck constant and $\varepsilon_0$ the permittivity of vacuum. In free space, the photonic LDOS is $\rho_0 = \omega^2/(\pi^2 c^3)$[26]. Thus, $\rho_{\mathbf{n}}(\mathbf{r})/\rho_0$ is the nearfield enhancement of the plasmonic nanoparticle. Below, we will display the photonic LDOS $\rho_{\mathbf{n}}(\mathbf{r})$ in 3D as well as the averaged LDOS (obtained by averaging $\rho_{\mathbf{n}}$ over all possible dipole orientations) at specific two-dimensional (2D) slices through the volume. Throughout the LDOS maps are scaled to the respective maxima.

Figure 3 and Supplementary Movie 1 show the photonic LDOS for the dipolar and quadrupolar modes of the nanocuboid as reconstructed from the experimental EELS data. For comparison, we also show the simulated photonic LDOS, which is obtained from simulations using as input the particle geometry only. The different panels of the figure report a 3D view of the photonic LDOS, including the directional information discussed above, as well as slice projections of the averaged LDOS at different planes above and below the particle. We observe excellent agreement between the experimental and simulated LDOS maps, demonstrating the applicability of our reconstruction scheme. The projected LDOS follows the electric nearfield distribution of the dipolar and quadrupolar plasmon modes, however, without any information about forward or backward direction because $\rho_{\mathbf{n}}(\mathbf{r})$ is averaged over the oscillation period of the fields.

The reconstructed photonic LDOS of the coupled nanodisks is shown in Fig. 4 and Supplementary Movie 2 for the five most prominent loss peaks in the EELS spectra (Fig. 2e). Mode (a) is

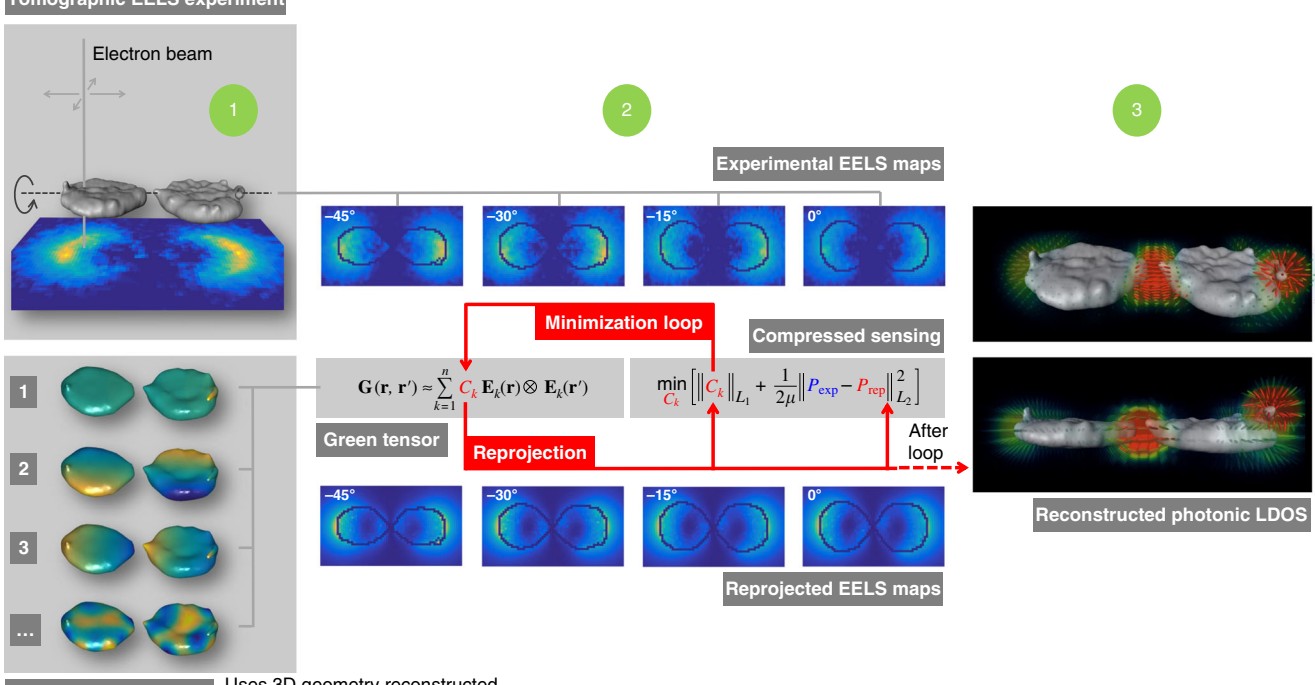

**Fig. 1** Tomography scheme for 3D photonic LDOS reconstruction. In a first step we measure high-angle annular dark-field (HAADF) images and electron energy-loss spectroscopy (EELS) maps for different tilt angles. From the HAADF data, we reconstruct the particle geometry and use the geometry for the computation of a generic eigenmode basis $\mathbf{E}_k(\mathbf{r})$. In a second step, the Green tensor $\mathbf{G}(\mathbf{r}, \mathbf{r}')$ is decomposed into these eigenmodes and the reprojected EELS maps are computed using some initial guess for the expansion coefficients $C_k$. These coefficients are determined through solution of an inverse problem such that the difference between the measured and reprojected EELS maps becomes minimized, with a compressed sensing bias that favors decompositions with as few eigenmodes as possible. After minimization, in the third step we use the reconstructed Green tensor to visualize the photonic environment

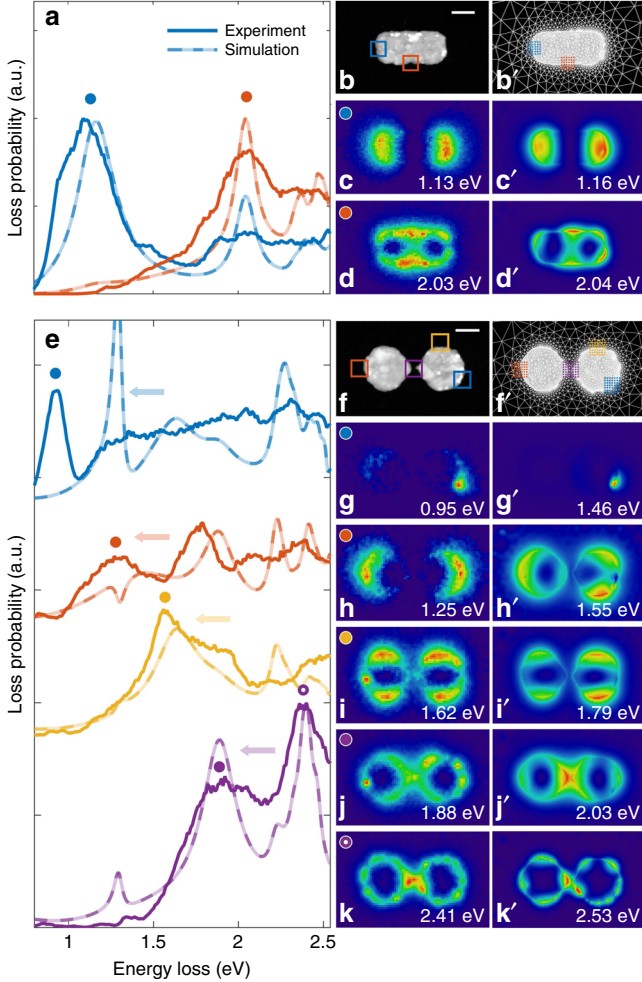

**Fig. 2** Selected EELS spectra and maps for nanocuboid and coupled nanodisks. **a** Measured and simulated EELS spectra for nanocuboid. The *red* and *blue squares* in **b** indicate the regions over which spectra are averaged. **c** and **d** EELS maps of the dipolar and quadrupolar modes at the resonance energies indicated by symbols in **a**. **b′-d′** Same as **b-d**, but for simulation results. **e-k′** Report results for the coupled nanodisks. Simulated spectra in **e** are shifted by 0.2 eV to lower energies (see *arrow*) to achieve better agreement for the modes at higher loss energies. The scale bar is 100 nm

strongly localized around the protrusion on the right particle, which is probably an artifact caused by an incomplete lift-off during electron-beam lithography fabrication. There is additionally a small but noticeable LDOS enhancement in the gap region that we interpret in terms of a hybridization between the localized mode and the bonding dipolar mode (b). For mode (b) the dipole moments of the individual particles are parallely aligned along the $x$ direction. This leads to a strong nearfield and LDOS enhancement in the gap region, a so-called hot spot, which gives no significant signal in 2D EELS maps[11] but can be clearly resolved in our tomographic reconstruction. For modes (c) and (d), the dipole moments of the individual particles are aligned parellely and antiparallely in the $y$ direction, respectively, whereas for the antibonding mode (e), the individual dipole moments are aligned antiparallely in the $x$ direction. As can be clearly seen in the right figure of panel (e), in the gap region the LDOS becomes strongly reduced and the electric nearfields point into the $z$ direction.

## Discussion

The comparison between the experimental and simulated EELS spectra in Fig. 2e is not perfect, although the reprojected EELS maps agree well with the measured ones (Supplementary Fig. 4). This demonstrates that our tomography approach is of general nature and that the plasmonic eigenmodes provide only a generic basis, without any significant bias in the reconstruction. The reason for the differences between experiment and simulation is not yet fully clear. The systematic energy shift of about 0.2 eV is in accordance to previous work[24] and is most likely due to a different metallic dielectric function with respect to the one used in simulations, caused by silver being not monocrystalline and possible chemical changes between deposition and TEM experiments. Also the thin carbon layer, which is deposited on the samples is not taken into acccount in the simulations and may cause a shift. An additional energy shift is observed during the experiments, which is most likely caused by carbon contamination (Supplementary Figs. 5–7). In the simulated spectra of Fig. 2e, we observe for the protrusion mode a dip, reminiscent of a Fano lineshape[34], caused by the interaction with the energetically close bonding mode. As a side effect, the loss peak of the bonding mode is only faint, in contrast to experiment where the peak is clearly visible. Finally, in the experimental EELS maps we observe an additional small protrusion on the left particle that

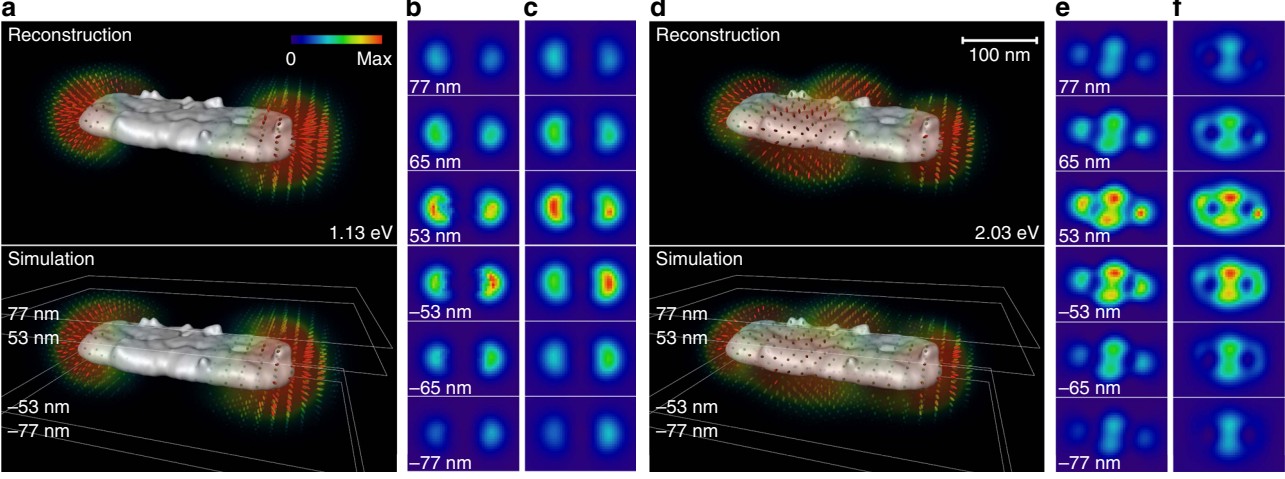

**Fig. 3** Reconstructed photonic LDOS for nanocuboid. **a** Reconstructed and simulated photonic LDOS for the dipole mode of the nanocuboid. The color of the pencils represents the LDOS magnitude, the orientation indicates the direction **n** along which $\rho_\mathbf{n}(\mathbf{r})$ is maximal. We additionally show the **b** reconstructed and **c** simulated averaged LDOS in different layers above and below the nanocuboid. **d-f** Same as **a-c** but for the quadrupole mode

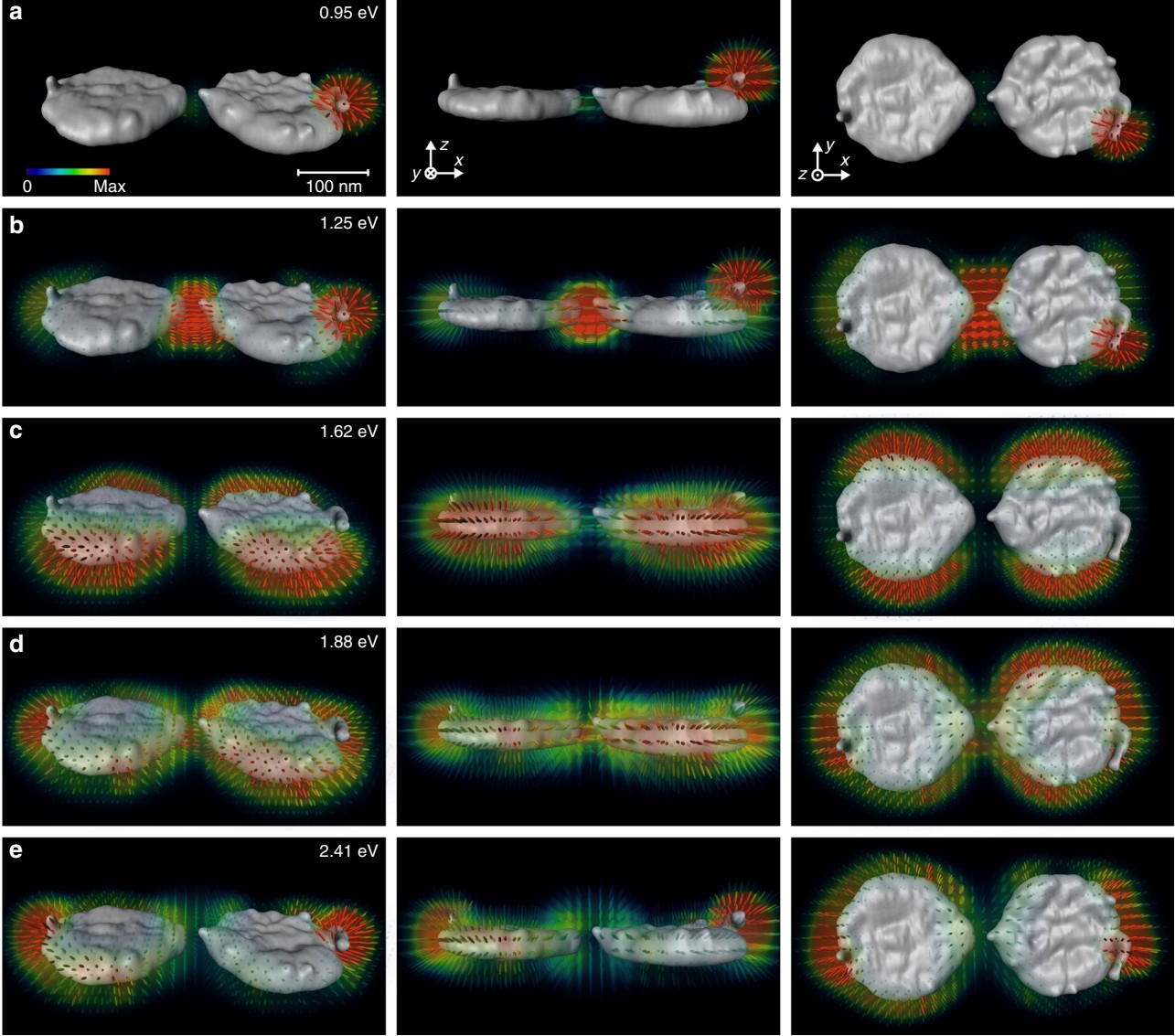

**Fig. 4** Reconstructed LDOS for coupled disks. The reconstruction is done for the five main peaks shown in Fig. 2e. The three columns show the LDOS from different viewing angles. **a** Localized plasmon mode at protrusion of right particle. For the other modes, the dipole moments of the disks are coupled in order of **b** → →, **c** ↑↓, **d** ↑↑ and **e** →←. Mode **d** exhibits an additional hybridization with the quadrupolar disk modes

does not appear in the simulated and reprojected maps (Fig. 2 and Supplementary Fig. 4). This deficiency might be due to differences between the true particle morphology and the one used in simulations and reprojections, or a too small eigenmode basis.

As possible extensions of our work, an additional feedback loop between the simulation input, including particle geometry and dielectric functions, and the EELS sinograms could allow for geometry optimizations or dielectric function extractions. Preliminary theoretical work[17] suggests that the tomographic reconstruction could be done with a significantly smaller number of measurement points, which would allow to diminish contamination-induced peak shifts during data acquisition.

In conclusion, we have experimentally demonstrated a tomographic approach for reconstructing the complete photonic environment of plasmonic nanoparticles. Our approach is generally applicable to dielectric or metallic nanostructures, regardless of size and complexity. Future work will investigate structures consisting of composite materials and the coupling effects between plasmonic nanoparticles and quantum emitters, such as semiconductor quantum dots. The capability of mapping

the three-dimensional photonic environment with nanometre resolution is expected to be of great interest for a large variety of nanophotonics systems, such as dielectric spheres or localized cavity modes in photonic crystals.

## Methods

**Sample preparation**. Electron beam lithography in a RAITH e-line system was applied to produce silver nanostructures of 30 nm thickness on a 5 nm $Si_3N_4$ membrane (TEMwindows) using a standard lift-off process with a poly(methyl-metacrylate) resist[9]. Grids with two rectangular $Si_3N_4$ windows (1.5 mm × 100 μm) were chosen to allow imaging under large tilt angles. Several structures of different shapes and sizes were designed. The structures chosen for the experiments were nanocuboids with lateral dimensions of 300 nm × 140 nm and coupled disks with diameters of 180 nm and a gap distance of 25 nm. The nominal height of all particles was 30 nm. To eliminate charging during the TEM measurements, a thin carbon layer (~1–2 nm) was sputtered on the sample (EPA 100, Leybold-Heraeus GmbH (Germany)). Without deposition of a carbon layer, movement of samples was observed in the TEM at large tilt angles.

**EELS acquisition**. Experiments were performed on a $C_S$-probe-corrected FEI Titan[3] 60–300 microscope, with an X-FEG field-emission electron gun and a Wien-type monochromator operated in decelerated gun-lens mode. The

acceleration voltage was 300 kV and excitation of the monochromator was set to 1.0. EELS spectra were acquired on a Gatan Imaging Filter (GIF Quantum) in hardware-synchronized mode without binning. Energy spread of the monochromated beam was measured to 150 meV as full-width at half-maximum of the zero-loss peak. The beam convergence semi-angle was 15 mrad and the collection semi-angle was 20.5 mrad. All spectrum images (SIs) were acquired with a size of $82 \times 52$ pixels at a pixel size of 7.7 nm with an exposure time of 5 ms per pixel. The total acquisition time for each spectrum image was 100 s. HAADF STEM survey images were acquired before SI acquisition with a pixel size of 1 nm at $1024 \times 1024$ pixels.

**EELS data processing**. All spectrum images were aligned in energy by shifting spectra relative to the positions of the zero-loss peak. After energy-shift alignment the spectra were normalized to the maximum intensity of the zero-loss peak and deconvolved using Richardson–Lucy deconvolution with 25 iterations in a home-made analysis program for spectrum images[36]. Spectra in the four corners of the spectrum images were averaged over $10 \times 10$ pixels per corner and were used as point-spread function for deconvolution. Supplementary Figs. 1 and 2 show the effects of the Richardson–Lucy deconvolution for spectra and resonance maps, respectively. In the spectra the contribution of the zero-loss peak is well removed and peaks are sharpened, while noise increases in the deconvolved data set. For resonance maps, little difference is visible before and after deconvolution, though removal of the background facilitates reconstruction. To extract 2D maps of surface plasmon resonances, the spectra were integrated over an energy region of 0.19 eV around the peak in the deconvolved spectrum images.

**HAADF STEM and EELS SI tilt series acquisition**. For the experiments, we used a Fischione 2020 tomography sample holder. For the nanocuboid sample tilt series were acquired between −75° and + 72° with steps of 10° from −70° to + 70° and two additional tilt angles at −75° and +72°, for the coupled disks the tilt range went from −75° to + 69° with steps of 15° from −75° to + 60° and two additional tilt angles at + 65° and + 69°. The additional tilt angles were used to maximize the tilt range at the highest possible sample tilt before shadowing of the sample. EELS SIs at each tilt angle were acquired and processed as described above. For the HAADF STEM reconstruction the survey images were used and rebinned by a factor of 2 to a pixel size of 2 nm.

**Tilt series alignment**. Alignment of the tomographic tilt series was done using center of mass methods based on the HAADF STEM survey images acquired just before EELS SI acquisition[24]. The projections were aligned in x-direction and y-direction by calculating their center of mass. The tilt axis was found by calculating rotational centers from sinograms, based on the center of mass. The calculated alignment parameters were used to align both the HAADF STEM tilt series and the tilt series of EELS spectrum images. To compensate for the possible drift between the survey images and SI acquisition, the SIs were aligned to the corresponding survey image by cross-correlation between the survey image and the HAADF signal acquired during SI acquisition.

**Reconstruction of the HAADF STEM tilt series**. The HAADF STEM tilt series was reconstructed using a total-variation (TV) minimization algorithm using the full 3D gradient calculation as described in previous work[31]. The normalization parameter $\mu$ was set to $2^6$. The HAADF STEM reconstructions were segmented using an Otsu threshold. From the segmented reconstruction, a triangulation of the silver surface was calculated, which was used as input for both simulations and reconstruction of particle plasmon fields.

**Reconstruction of photonic LDOS**. The reconstruction of the photonic LDOS was performed as described in ref. 17. As input geometry, reconstructions of the surface from HAADF STEM tilt series were used. The input geometry was aligned in x-direction and y-direction to correspond to the measured EELS maps. As an impact, parameters for the evaluation of the trajectory integrals pixel coordinates from the experimental EELS maps were used.

The compressed sensing minimization was performed with a L1/L2 optimization[28] using the toolbox YALL1 (online at http://yall1.blogs.rice.edu). The mixing parameter was set to $\mu = 0.95$. With the latter choice, we achieved good agreement between the measured and backprojected maps (Supplementary Figs. 3 and 4), while still introducing sufficient compressed-sensing bias for a restricted eigenmode basis.

For the computation of the eigenmodes, we used surface reconstructions from HAADF STEM tilt series as described above. The number of triangulation elements was 1700 for the cuboid structure and 2700 for the coupled disks. The eigenmodes were obtained by computing the eigenvalues and eigenvectors of the $\Sigma$ matrix from a boundary element approach, see Eq. (21) of ref. 35, at the (real) frequencies of the plasmon resonances extracted from the simulated EELS spectra. For the cuboid structure, we also varied the frequencies for the eigenmode evaluation by ±0.2 eV, but did not find any significant changes of the reconstructed quantities. For the reconstruction procedure, we used the 50 eigenmodes of lowest energy. Consideration of more eigenmodes (up to 100) did not noticeably change the reconstructed quantities. Our compressed sensing optimization also turned out to

be extremely robust with respect to the number of measurement points, as well as to other parameters such as $\mu$.

For the dipolar and quadrupolar cuboid modes, we used EELS measurement data for angles from −50° to 50°. Because our BEM approach uses boundary elements of finite size, fields close to the boundary may suffer from discretization errors and we thus discarded in the reconstruction procedure impact parameters that were closer than 5 nm to the nanoparticle. Simulations and reprojections of EELS maps are compared with EELS measurements in Supplementary Fig. 3. The photonic LDOS of the coupled disks was reconstructed for five modes. For mode (a) of lowest energy, we used experimental EELS maps for tilt angles from −60° to 0°. EELS maps for larger tilt angles were not taken into account because of the contamination-induced red-shift during data acquisition. The remaining four modes were reconstructed from EELS maps for tilt angles −60° to 60°, −45° to 45°, −45° to 45° and −60° to 30°, respectively. Measured EELS maps and reprojections for different tilt angles are shown in Supplementary Fig. 4.

**Simulation of EELS spectra and maps**. The electron energy-loss simulations for the two geometries were carried out with the MNPBEM-Toolbox[17, 32, 37]. For simulations of EELS spectra, the same simplified surface reconstruction as for the calculation of eigenmodes was used. To account for the 5 nm thick $Si_3N_4$ membrane used in experiment, we added a 5 nm thick layer with a permittivity of 4 at the bottom of the particles. Together with the layer structure, the overall number of triangulation elements was 6000 and 8000 for cuboid and coupled disks, respectively. A dielectric function obtained from optical experiments[33] was used for both geometries. The reprojected maps were computed without consideration of the thin substrate.

**Data availability**. The data sets generated during and/or analysed during the current study are available from the corresponding author on reasonable request.

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

## Acknowledgements

We are grateful to Joachim Krenn and Harald Ditlbacher for providing access to electron beam lithography and for helpful discussion. This work has been supported in part by the Austrian Science Fund FWF under project P27299-N27 and the SFB F49 NextLite (F4906-N23), by NAWI Graz, and by the European Union within the 7th Framework Program (FP7/2007–2013) under Grant Agreement no. 312483 (ESTEEM2).

## Author contributions

F.-P.S. prepared the samples, G.H. performed TEM experiments, EELS data processing and reconstruction of the HAADF STEM data. A.H. and A.T. performed reconstruction of the photonic LDOS and simulations. All authors participated in the interpretation of the data and the data analysis, and helped in editing the manuscript.

## Additional information

**Competing interests:** The authors declare no competing financial interests.

