## [Peer Review file · Nature Communications]

Reviewers' comments:

Reviewer #1 (Remarks to the Author):

Hörl and co-workers present an outstanding work on the reconstruction of the LDOS associated with plasmonic modes near metallic nanostructures. They use a combination of recently developed theoretical methods along with state-of-the-art electron microscopy techniques (tomography and loss spectroscopy). The agreement that they find between the simulated LDOS (via solution of the EM problem for the geometry derived from tomographic reconstruction of the structure) and the LDOS retrieved from experiment is remarkable. The paper is clearly written and the techniques are clearly explained and referenced. In fact, this work really pushes the field well beyond its current state and I anticipate that it will have large impact in the electron microscopy and plasmonics communities alike.

I only have a few questions that the authors might want to address before publishing their work:

1) The validity of of Eq. (1) should be further explained, in particular the reason why the eigenmodes seem to be assumed to be real. This point is also discussed in the Methods of Ref. 17. Are the particles sufficiently small as to neglect retardation? Can the error produced by this approximation be quantified?

2) Can the methods be readily applied to nonmetallic structures (e.g., Mie modes) and/or extended systems (photonic crystals, plasmonic crystals)?

3) It would be useful to have an analysis of the error in the reconstruction of the LDOS. Is there a way to do this in a simple way (e.g., by studying the effect of randomly displacing the reconstructed value in a point-by-point fashion)?

4) Additionally, it would be interesting to see how uncertainties in the EELS measurements (e.g., from the acquisition statistics, non-unity spectral weight imposed by the experimental system, background subtraction, etc.) translate into an error in the reconstructed modes, and how this can possibly restrict the photonic/plasmonic structures for which the technique can be more suited.

5) The authors claim that even fewer experimental points should be sufficient to provide a decent reconstruction. A few words on the number of data points needed to achieve a certain level of precision in the reconstruction would be equally useful.

Reviewer #2 (Remarks to the Author):

Comments to the manuscript: NCOMMS-16-24742

GENERAL OVERVIEW

The paper "Tomographic reconstruction of the photonic environment of plasmonic nanoparticles" introduces a tomographic scheme for retrieving the 3D photonic LDOS of plasmonic nanoparticles. The authors demonstrate how to calculate the 3D Green tensor of plasmonic nanoparticles from experimental measurements using transmission electron microscopy.

The paper builds up on methods that the authors have been publishing separately in previous papers:

- First, the authors recover the shape of the particles using HAADF STEM tomography and:

1. 3D Eigenmodes $E_k(r)$ of different energies are calculated from the shape of the particles considering retardation effects.

2. 2D EELS maps are also simulated at the same energies, and from different orientations. These procedures have already been presented by the authors in [Hörl et al. PRL 111, 2013] and in [Haberfehlner et al. Nano Lett. 2015, 15].

- Then they use a minimisation procedure (within the paradigm of compressed sensing) using the experimental data to find the optimal basis of eigenmodes AND their coefficients C_k that better describe the dyadic Green tensor (explained in [Hörl et al., ACS Photonics 2015,2] but using simulated data).
- Finally, from 3D Green tensor the photonic LDOS in three-dimensions is computed.

I have not big concerns with the theoretical foundations of the methodology used in the paper. The authors are well-known experts in the field, and most of the methods have been reported previously in peer-reviewed high-impact journals.

As such, the paper is like an end-of-story paper that combines all the previous protocols presented in the past using real data: (i) the shape of the particle and (ii) a tomographic series of EELS maps.

The method is powerful because is applicable to larger particles with general geometries that may not be in the quasistatic regime.

My only concern is on one detail of the experimental side of the draft, that the authors mention. In particular, the observation of some energy shifts on the plasmon resonances.

ENERGY SHIFTS

Although it does not detract the full procedure described in the paper, the authors mention the observation of a red-shift of about 0.2 eV of the resonances with increasing tilt angles. The authors assume that is due to carbon contamination and they correct it simply shifting all back the resonances by about 0.2 degrees.

In page 7, line 9 they mention that "the reason for the differences between experiment and simulation is not yet fully clear". "... the energy shift is in accordance to previous work and might be due to a different metallic dielectric function with respect to the one used in the simulations, or to energy shifts caused by contamination". This is coherent with a previous work of the authors [Haberfehlner et al. Nano Lett. 2015,15] where they explained the shifts like being "most likely due to nanoparticle aging and the modification of grain sizes, which leads to a modification of the metal permittivity and in turn to a red-shift of the plasmonic resonances".

That would agree also with the HAADF images of the particles presented in the draft which have contrast features that indicate that the particles are not single crystalline.

But then, in the supporting Information, Point 1, the authors state that "During acquisition of the tilt series, carbon contamination slowly built up... causing continuous shifts of the resonance peaks".

-Then, what is the origin of the shift ¿carbon contamination or particle crystallinity?

The authors deposited a film of carbon on the sample to reduce charging effects. Si_3N_4 is an insulator, and although for the thickness of the substrate and the acceleration used it is unlikely, it is also true that due to the long exposure time used in the experiments, uncompensated positive electric charge might accumulate on the sample.

-¿Did the authors observe any charging of the sample, for example using electron holography?

-I think that it would be easy for the authors to test experimentally if contamination is originating the red shift. They simply have to plasma clean the samples, to go to the microscope and to acquire an EELS map at a high-tilt angle. If the red-shift is still measured, then contamination is not the origin of the shift.

Moreover, the authors also mention that although they observed energy shifts, "the modal distribution in the plasmon resonance map did not change significantly". This is an interesting observation because may retardation effects and not contamination may be the cause of the red-shift. In general retardation fields affects the energy and the amplitude of a resonant mode but not its projected symmetry. At high tilts, like in the experiments performed here the corresponding resonant fields have more coupling to the Si₃N₄ substrate and could red-shift. Directionality in plasmonics is important [Coenen et al. Nano Lett 2011, 11]. Also the substrate should break z symmetry, and mismatches between reprojections and experimental maps may be larger as trajectories become more parallel to the substrate [Nicoletti et al. Nature 2013].

OTHER COMMENTS

-The paper is well-written with a solid background that has been well referenced. But the paper lacks some reference to other techniques for measuring plasmons like cathodoluminescence (CL). CL is being successfully applied in electron microscopy [Gómez-Medina et al. New Journal of Physics 10, 2008; Zagonel et al. Nano Lett 11, 2011]. CL has better energy resolution than EELS although is less sensitive to HO modes. In [Losquin et al. 2015 Nano Lett] both techniques are compared.

-For the acquisition of the HAADF tilt series, why for the cuboid and the disks you acquired "additional images" at -75 and +72 and at +65 and +69 degrees?

-¿What do you meant when in the paragraph "Reconstruction of the Photonic LDOS" , in page 10, line 19 you mention that "EELS measurements that where closer to 5 nm to the nanoparticle were discarded"?

-Page 3, line 4, "...Unfortunately, these approaches come along with several shortcomings..." ¿Can the authors explicit this sentence a bit more?

-I missed scale bars on the images.

-EELS maps could have some annotation added, for example their energy

- Just, for curiosity, have the authors explored if the solution of the Green function obtained with the minimization method is very sensitive to the value of μ ?

Reviewer #3 (Remarks to the Author):

Dear Editor,

The authors are reporting a "tomographic reconstruction" technique to visualize the photonic/plasmonic fields of nanostructures using electron energy loss spectroscopy where the fields are excited and probed simultaneously by a fast electron. To achieve the 3D visualization, they introduce a novel tomography scheme, as they claim.

The manuscript has one major issue pertaining to the idea of obtaining the 3D fields (or LDOS): The way the 3D spatial distribution is obtained is actually not through "reconstruction" or "tomography". Their procedure is, in simple words, to fit the already simulated results to the experimental data at certain points in space. And then to "extrapolate" these fitting results (using the already calculated eigenstates) beyond what experiments are actually measuring. To further elaborate:

1. Why is this called a reconstruction? The final 3D distribution is not directly obtained from the experimental data. For instance, in Fig.1, there is no any experimental signal between the discs (see EELS maps at -45,-30,-15, 0 degree tilts). But, in the reconstructed 3D image the fields are significant in the gap. How is it possible to "reconstruct" a quantity from non-existing quantities?

Similarly, fast electrons probe the electric field component that is only parallel to their propagation direction. With the tilt axis shown in Figure 1, the probing electrons never interact with the transverse component of the field. But, in the final reconstruction this transverse component is present, or can be computed. Again, how is it possible to obtain a quantity that is not present in the experimental data?

2. What is point of tilting the specimen? What new information does this tilting provide besides improving the fitting parameters? There is no any indication in the manuscript that these projections are actually used to reconstruct the 3D distribution.

In a work published few years ago (*Nature*, 502, 80, 2013), plasmonic modes of a nanocube were indeed reconstructed by using only the experimental tilt series as a starting point, with no modeling or eigenstate calculations required. This is what tomographic reconstruction means.

Without these points clarified by the authors, I cannot recommend the publication of the manuscript. In my opinion, the current work uses modeling so heavily that there is little to none experimental data/information in the final 3D plasmonic maps.

Response to Reviewer:

We thank all Reviewers for their valuable reports and for pointing out where further information is needed in our manuscript. We understand that Reviewers #1 and #2 support publication but have raised a number of technical points that should be clarified before publication. In our detailed response to the Reviewers, given below, we address point by point the questions of the Reviewers. A number of changes have been introduced in our revised manuscript.

As regarding Reviewer #3, it appears that he/she has not fully appreciated the primary rationale behind our approach. In our reply, given below, we argue why we believe that our approach is much more than a simple fitting procedure and why our scheme significantly improves on previous attempts. It might be that we have not been overly clear regarding the general philosophy in our first submission, and we have thus expanded the discussion in our revised manuscript.

We hope that with the changes made in the revised manuscript and the point by point consideration of all questions raised in the reports, the Reviewers can now recommend publication of our manuscript in Nature Communications.

Reviewers' comments:

Reviewer #1 (Remarks to the Author):

Hörl and co-workers present an outstanding work on the reconstruction of the LDOS associated with plasmonic modes near metallic nanostructures. They use a combination of recently developed theoretical methods along with state-of-the-art electron microscopy techniques (tomography and loss spectroscopy). The agreement that they find between the simulated LDOS (via solution of the EM problem for the geometry derived from tomographic reconstruction of the structure) and the LDOS retrieved from experiment is remarkable. The paper is clearly written and the techniques are clearly explained and referenced. In fact, this work really pushes the field well beyond its current state and I anticipate that it will have large impact in the electron microscopy and plasmonics communities alike.

I only have a few questions that the authors might want to address before publishing their work:

1) The validity of of Eq. (1) should be further explained, in particular the reason why the eigenmodes seem to be assumed to be real. This point is also discussed in the Methods of Ref. 17. Are the particles sufficiently small as to neglect retardation? Can the error produced by this approximation be quantified?

Our response:

The eigenmodes $E_k(r)$ are complex-valued, the expansion of Eq. (1) is the general decomposition of the retarded Green tensor in terms of eigenmodes, see for instance Eq. (9) of C. Sauvan et al., "Modal representation of spatial coherence in dissipative and resonant photonic systems", Phys. Rev. A 89, 043825 (2014). Importantly, there is no complex conjugate or similar involved. As Reviewer #3 had

similar problems regarding this point, we suppose that our original discussion has not been sufficiently clear and we have tried to improve our presentation in the revised manuscript.

Actions taken:

A new paragraph discussing this issue has been introduced at the end of the section “Results > Tomography scheme”.

2) Can the methods be readily applied to nonmetallic structures (e.g., Mie modes) and/or extended systems (photonic crystals, plasmonic crystals)?

Our response:

Quite generally, our approach works for any system whose photonic environment can be approximated by the Green tensor decomposition of Eq. (1) with a finite number of eigenmodes. We expect that a truncated expansion basis works best for systems exhibiting pronounced resonances in the electromagnetic response, which is certainly the case for dielectric spheres or localized cavity modes in a photonic crystal. Things might be more difficult for extended systems because of the continuum of eigenmodes, although such systems might be interesting for future investigations.

Actions taken:

We have included a sentence in the conclusion: “The capability of mapping the three-dimensional photonic environment with nanometer resolution is expected to be of great interest for a large variety of nanophotonics systems, such as dielectric spheres or localized cavity modes in photonic crystals.”

3) It would be useful to have an analysis of the error in the reconstruction of the LDOS. Is there a way to do this in a simple way (e.g., by studying the effect of randomly displacing the reconstructed value in a point-by-point fashion)?

Our response:

The referee poses a question that is difficult to answer. First, we can test the quality of our expansion by comparing the measured EELS maps with the reprojected ones, as shown in Figs. S3 and S4. There we find extremely good agreement, which indicates a high quality of our reconstruction process (we could even quantify the small error between the experimental and reprojected maps). However, to the best of our knowledge there is no way to quantify the error for other reconstructed quantities, such as the photonic LDOS, except through comparison with complementary LDOS measurement results (which are very hard to obtain) or simulation results, as we do in this work.

This problem seems to be inherent to practically all tomography schemes, even for the usual computerized tomography approach: while a complete reconstruction is in principle possible using the inverse Radon transformation, any loss of information, such as caused by the notorious missing wedge problem, introduces an uncontrollable element to the reconstruction process. Although filtered back projection is known to work perfectly in such situations, it is no longer possible to quantify the error for the reconstructed quantities. The robustness of the reconstruction procedure with respect to the measurement results (e.g. checking whether one obtains similar results when using less tilt angles or measurement points in the reconstruction) usually indicates whether a reliable reconstruction is

possibly or not, but it is never possible to exclude possible artifacts caused by the incomplete measurement data.

4) Additionally, it would be interesting to see how uncertainties in the EELS measurements (e.g., from the acquisition statistics, non-unity spectral weight imposed by the experimental system, background subtraction, etc.) translate into an error in the reconstructed modes, and how this can possibly restrict the photonic/plasmonic structures for which the technique can be more suited.

Our response:

As described above, estimation of the error in the reconstruction can be hardly quantified, but what can be estimated is the reliability of the projection data, which serves as basis for the reconstruction

To estimate the signal to noise ratio in the projections we can adapt Eq. (5.23) from Egerton, Electron Energy-Loss Spectroscopy in the Electron Microscope (Springer 2011). Assuming Poisson noise the SNR is defined as

$$SNR = DQE^{1/2} \frac{I_S}{(I_S + hI_B)^{1/2}}$$

DQE is the detector quantum efficiency, which is 0.3 for our detector (see Riegler, Kothleitner: "EELS detection limits revisited: Ruby — a case study", Ultramicroscopy 110, 1004-1013, 2010). h is the statistical error from background (or in our case zero-loss) subtraction. For core-loss analysis h is typically 5-10, but we expect lower values for the low-loss region due to the high signal. I_B is the background (zero-loss) intensity and I_S is the intensity of the signal after zero-loss removal. For our analysis both are integrated over a range of 0.2 eV, the same value used for the extraction of the EELS maps in the experiment.

We next calculate the SNR for the cuboid sample for the dipolar and quadrupolar modes for projections at 0° tilt and at -75° tilt, and vary h between 1 and 10 to estimate the impact of the background subtraction on the SNR. As shown in the figure below, for $h = 1$ the SNR goes up to 30 and even for $h = 10$ the SNR gets as high as 10, and is well above the critical threshold of 3 (98% certainty of detection) for all relevant locations. We therefore conclude that the data are certainly good enough for a reliable reconstruction. It can be additionally observed that the SNR is higher for electron trajectories outside the particles, which has to do with the reduction of the signal due to concurrent scattering events. This effect is present especially at large tilt angles, where the effective thickness of the sample is increased due to sample tilt.

Calculated SNR maps for the dipolar mode at (a) 0° tilt and (b) -75° tilt and for the quadupolar mode at (c) 0° tilt and (d) -75° tilt. The colorbars are shown for different values of h .

5) The authors claim that even fewer experimental points should be sufficient to provide a decent reconstruction. A few words on the number of data points needed to achieve a certain level of precision in the reconstruction would be equally useful.

Our response:

We have tried to address this point in Ref. 17 where we show in Fig. 3 that reconstruction is even possible for a strongly reduced data set. As a rule of thumb, the number of data points should be such that a further reduction of say 25%, e.g. through coarse graining, still gives similar reconstruction results. Another constraint is given by the expansion coefficients C_k in Eq. (1): apparently the number of measurement points should be much larger than the number of coefficients. A more general discussion regarding the adequate number of measurement points can be found in the literature on compressed sensing, see e.g. Refs. [22,23].

Actions taken:

This issue is now briefly addressed in Methods > Reconstruction of photonic LDOS.

Reviewer #2 (Remarks to the Author):

Comments to the manuscript: NCOMMS-16-24742

GENERAL OVERVIEW

The paper "Tomographic reconstruction of the photonic environment of plasmonic nanoparticles" introduces a tomographic scheme for retrieving the 3D photonic LDOS of plasmonic nanoparticles. The authors demonstrate how to calculate the 3D Green tensor of plasmonic nanoparticles from experimental measurements using transmission electron microscopy.

The paper builds up on methods that the authors have been publishing separately in previous papers:

- First, the authors recover the shape of the particles using HAADF STEM tomography and:
 1. 3D Eigenmodes $E_k(r)$ of different energies are calculated from the shape of the particles considering retardation effects.
 2. 2D EELS maps are also simulated at the same energies, and from different orientations. These procedures have already been presented by the authors in [Hörl et al. PRL 111, 2013] and in [Haberfehlner et al. Nano Lett. 2015, 15].
- Then they use a minimisation procedure (within the paradigm of compressed sensing) using the experimental data to find the optimal basis of eigenmodes AND their coefficients C_k that better describe the dyadic Green tensor (explained in [Hörl et al., ACS Photonics 2015,2] but using simulated data).
- Finally, from 3D Green tensor the photonic LDOS in three-dimensions is computed.

I have not big concerns with the theoretical foundations of the methodology used in the paper. The authors are well-known experts in the field, and most of the methods have been reported previously in peer-reviewed high-impact journals.

As such, the paper is like an end-of-story paper that combines all the previous protocols presented in the past using real data: (i) the shape of the particle and (ii) a tomographic series of EELS maps.

The method is powerful because is applicable to larger particles with general geometries that may not be in the quasistatic regime.

My only concern is on one detail of the experimental side of the draft, that the authors mention. In particular, the observation of some energy shifts on the plasmon resonances.

ENERGY SHIFTS

Although it does not detract the full procedure described in the paper, the authors mention the observation of a red-shift of about 0.2 eV of the resonances with increasing tilt angles. The authors assume that is due to carbon contamination and they correct it simply shifting all back the resonances by about 0.2 degrees.

In page 7, line 9 they mention that “the reason for the differences between experiment and simulation is not yet fully clear”. “... the energy shift is in accordance to previous work and might be due to a different metallic dielectric function with respect to the one used in the simulations, or to energy shifts caused by contamination”. This is coherent with a previous work of the authors [Haberfehlner et al. Nano Lett. 2015,15] where they explained the shifts like being “most likely due

to nanoparticle aging and the modification of grain sizes, which leads to a modification of the metal permittivity and in turn to a red-shift of the plasmonic resonances”.

That would agree also with the HAADF images of the particles presented in the draft which have contrast features that indicate that the particles are not single crystalline.

But then, in the supporting Information, Point 1, the authors state that “During acquisition of the tilt series, carbon contamination slowly built up... causing continuous shifts of the resonance peaks”.

-Then, what is the origin of the shift ¿carbon contamination or particle crystallinity?

Our response:

Both cause a shift, but the crystallinity-induced shift is already present at the beginning of the experiment. In addition to the initially observed mismatch between simulated and experimental peak positions, which we attribute mostly to crystallinity and possibly also to changes of chemistry and the neglect of the thin deposited carbon layer in the simulations, we see during the data acquisition for different tilt angles an additional shift attributed to carbon contamination (Figs. S5 and S6).

Actions taken:

We clarified the two causes for the shift in the main text (Page 7, Par. 2) and added a comparison of HAADF STEM images before and after a tilt series, showing the contamination in the Supporting Information.

The authors deposited a film of carbon on the sample to reduce charging effects. Si₃N₄ is an insulator, and although for the thickness of the substrate and the acceleration used it is unlikely, it is also true that due to the long exposure time used in the experiments, uncompensated positive electric charge might accumulate on the sample.

-¿Did the authors observe any charging of the sample, for example using electron holography?

Our response:

The deposition of carbon was motivated by practical experience, aiming at stable samples under the electron beam. An uncoated tilted sample showed lateral shifts, resulting in strongly distorted STEM images. The shifts depend on scan speed, and it appeared as if the sample was pushed by the electron beam. With the deposition of a thin carbon layer the movement disappeared completely, which is why we attributed this effect to charging. Unfortunately, holography is not an available option on our microscope, but for further studies this is an interesting suggestion.

Actions taken:

We added a comment about sample movement in Methods > Sample preparation.

-I think that it would be easy for the authors to test experimentally if contamination is originating the red shift. They simply have to plasma clean the samples, to go to the microscope and to acquire an EELS map at a high-tilt angle. If the red-shift is still measured, then contamination is not the origin of the shift.

Our response:

Plasma cleaning was indeed performed after the experiment. The oxygen plasma however attacked the sample (at least as soon as the carbon was removed), leading to a change in morphology and chemistry of the silver nanoparticles. Whether a change in gas chemistry (reductive plasma) would improve this issue needs to be explored in a future study.

Moreover, the authors also mention that although they observed energy shifts, “the modal distribution in the plasmon resonance map did not change significantly”. This is an interesting observation because retardation effects and not contamination may be the cause of the red-shift. In general retardation fields affects the energy and the amplitude of a resonant mode but not its projected symmetry. At high tilts, like in the experiments performed here the corresponding resonant fields have more coupling to the Si₃N₄ substrate and could red-shift. Directionality in plasmonics is important [Coenen et al. Nano Lett 2011, 11]. Also the substrate should break z symmetry, and mismatches between reprojections and experimental maps may be larger as trajectories become more parallel to the substrate [Nicoletti et al. Nature 2013].

Our response:

This is an interesting idea and it would be great to be able to measure these direction-induced shifts. However, this requires pristine, unaltered samples to be able to exclude contamination effects. We cannot exclude directionality effects in our measurements, yet we are quite sure that contamination has a much larger impact due to the following reasons:

- *We compared peak position at 0° tilt angle before, during and after the experiment and observed the peak shift (Fig. S6). To clarify this, we have now indicated the peak energy in Fig. S6.*
- *The shift of the peak positions during acquisition (starting at a large negative angle to 0° to a large positive angle) appears quite linear (Fig. S5).*

OTHER COMMENTS

-The paper is well-written with a solid background that has been well referenced. But the paper lacks some reference to other techniques for measuring plasmons like cathodoluminescence (CL) .CL is being successfully applied in electron microscopy [Gómez-Medina et al. New Journal of Physics 10, 2008; Zagonel et al. Nano Lett 11, 2011]. CL has better energy resolution than EELS although is less sensitive to HO modes. In [Losquin et al. 2015 Nano lett] both techniques are compared.

Our response:

We thank the Referee for the suggestion to include also a brief discussion about cathodoluminescence.

Actions taken:

We have introduced the suggested references just before the Results section, and have additionally added the reference to the work of Atré et al. about CL tomography.

-For the acquisition of the HAADF tilt series, why for the cuboid and the disks you acquired “additional images” at -75 and +72 and at +65 and +69 degrees?

Our response:

For both samples (cuboid and disks) the tilt range was limited by shadowing from the TEM grid, which depends on the position of the sample on the grid. For the cuboid the sample was visible from -75° to +72° for the disks from -75° to +69°. For convenience we used a linear tilt scheme with “round” numbers and extended the tilt range we acquired with the “additional images” at the end of the tilt range.

Actions taken:

We added a sentence in Methods > HAADF STEM and EELS SI tilt series acquisition.

-¿What do you meant when in the paragraph “Reconstruction of the Photonic LDOS” , in page 10, line 19 you mention that “EELS measurements that where closer to 5 nm to the nanoparticle were discarded”?

We now write “Because our BEM approach uses boundary elements of finite size, fields close to the boundary may suffer from discretization errors and we thus discarded in the reconstruction procedure impact parameters that were closer than 5 nm to the nanoparticle.”

-Page 3, line 4, “...Unfortunately, these approaches come along with several shortcomings...” ¿Can the authors explicit this sentence a bit more?

We now write that the approaches “require a plasmonic response governed by a single mode and electron trajectories that do not penetrate the nanoparticle”. Other problems are sign changes in the eigenpotentials as well as extremely large fields of view to capture the long-range components of the plasmonic fields.

-I missed scale bars on the images.

Scale bars have been added in the revised manuscript.

-EELS maps could have some annotation added, for example their energy

We have annotated the EELS maps.

- Just, for curiosity, have the authors explored if the solution of the Green function obtained with the minimization method is very sensitive to the value of μ ?

The influence of μ in the compressed sensing optimization has been investigated. Indeed, we find that the entire optimization results are extremely robust with respect to variations of the input data (number of tilt angles and measurement points) as well as to variations of μ .

Reviewer #3 (Remarks to the Author):

Dear Editor,

The authors are reporting a “tomographic reconstruction” technique to visualize the photonic/plasmonic fields of nanostructures using electron energy loss spectroscopy where the fields

are excited and probed simultaneously by a fast electron. To achieve the 3D visualization, they introduce a novel tomography scheme, as they claim.

The manuscript has one major issue pertaining to the idea of obtaining the 3D fields (or LDOS): The way the 3D spatial distribution is obtained is actually not through “reconstruction” or “tomography”. Their procedure is, in simple words, to fit the already simulated results to the experimental data at certain points in space. And then to “extrapolate” these fitting results (using the already calculated eigenstates) beyond what experiments are actually measuring. To further elaborate:

1. Why is this called a reconstruction? The final 3D distribution is not directly obtained from the experimental data. For instance, in Fig.1, there is no any experimental signal between the discs (see EELS maps at -45,-30,-15, 0 degree tilts). But, in the reconstructed 3D image the fields are significant in the gap. How is it possible to “reconstruct” a quantity from non-existing quantities?

Our response:

First, we agree with the Referee that it would be nice to have a tomography scheme that works without any pre-knowledge, such as the Radon transformation or back-projection approach used by the Midgley group [15]. Unfortunately, this approach is valid only under extremely restrictive assumptions, such as the applicability of the quasistatic approximation or a plasmonic response governed by a single mode [14,17], and despite the beautiful results of the Midgley group for nanocubes we are not aware of any follow-up work (on the contrary, in [16] the same authors now use a different reconstruction scheme). For the general retarded case there does not exist a scalar quantity in 3D which can be linked to EELS maps by a Radon transformation, which is why we resorted to setting up and solving a more general inverse problem.

For the above reasons, in our approach we use some pre-knowledge (as detailed below), however, the implications are much less severe as envisioned by the Reviewer. Consider first a nanosphere, which is the only nanoplasmonic system that has an analytic solution: within Mie theory the vector spherical harmonics together with the spherical Bessel and Hankel functions provide the solutions of Maxwell's equations. On the other hand, these functions also provide a generic basis for arbitrary electromagnetic fields. Thus, outside the nanosphere any electromagnetic field can be written as a sum of such basis functions with properly chosen expansion coefficients. For the problem of our concern a huge number of coefficients would be needed in such an expansion, and the determination of the coefficients becomes unfeasible in practice.

In our work we thus employ a more educated guess for the basis functions and use the quasinormal modes for the particle geometry reconstructed tomographically from the HAADF tilt series. Again these modes serve as a basis for the electromagnetic fields, however, in contrast to Mie theory they are already well adapted to the problem under investigation, and in general a few of these modes suffice to properly expand the fields. This renders a reconstruction scheme (in terms of the solution of an inverse problem) feasible. Our compressed sensing optimization for obtaining the “best” coefficients is in accordance to related work, see Refs. [15,22-26].

We thus strongly disagree with the Reviewer that our approach is a simple “fit to the already simulated results”. Rather we provide a generic basis, which is chosen such that a small number of expansion coefficients allow representing the plasmonic modes, which makes our approach suitable for practical implementations.

Actions taken:

We have added at the end of the “Results > Tomography scheme” section a new paragraph where we explain the rationale behind our approach in more detail.

Similarly, fast electrons probe the electric field component that is only parallel to their propagation direction. With the tilt axis shown in Figure 1, the probing electrons never interact with the transverse component of the field. But, in the final reconstruction this transverse component is present, or can be computed. Again, how is it possible to obtain a quantity that is not present in the experimental data?

Our response:

As regarding the strong electric field in the hot spot region between the nanoparticles despite the missing EELS signal there, it is important to realize that we are not simply reconstructing the field vectors at all space points independently. Rather we use the pre-knowledge that the electromagnetic fields away from the nanoparticle boundary are solutions of the homogeneous Maxwell equations (as automatically built into the quasinormal modes), which poses severe constraints on possible field configurations.

By tilting the specimen and measuring the fields integrated along the electron trajectories (energy loss) we obtain a huge amount of (experimental!) constraints that must be fulfilled simultaneously in the determination of the expansion coefficients. Through this procedure we end up with a highly overdetermined problem where a few coefficients are obtained from a huge amount of experimental data. The high field strengths in the gap region turns out to be the only field configuration compatible with all measured data from the tilt series, and highlights the versatility of our approach.

We emphasize that the primary pre-knowledge entering our approach is the (reasonable) assumption that the plasmonic fields outside the nanoparticles are solutions of the homogeneous Maxwell's equations. The rest, namely the choice of an already well-adapted basis and the compressed sensing optimization for the determination of the expansion coefficients, is of rather technical nature and, in our view, does not represent a serious drawback for our approach.

2. What is point of tilting the specimen? What new information does this tilting provide besides improving the fitting parameters? There is no any indication in the manuscript that these projections are actually used to reconstruct the 3D distribution.

Our response:

Using a tilt series is inevitable for a proper reconstruction, more specifically, to obtain information about the integrated field values along different lines. While the necessity for using many tilt angles is apparent in the Radon transformation and the filtered back projection, it is less obvious for our solution of an inverse problem. However, we checked that our scheme fails when using a single EELS map due to the ambiguity of the EELS map with respect to different field configurations. Through tilting we obtain a certain amount of redundancy in our data which allows uniquely determining the proper plasmonic field distribution.

The experimental projections are used in the second term in Eq. (2). Here, for all tilt angles the difference between the experimental projections and the re-projections is minimized. A comparison is

given in Figs. S3 and S4 where we show that with a single (!) set of expansion coefficients C_k we can indeed reproduce the experimental maps for all (!) tilt angles.

In a work published few years ago (Nature, 502, 80, 2013), plasmonic modes of a nanocube were indeed reconstructed by using only the experimental tilt series as a starting point, with no modeling or eigenstate calculations required. This is what tomographic reconstruction means.

Our response:

As discussed above, despite the beautiful results of this work it is unclear what quantity is really reconstructed through the filtered back projection used by Nicoletti and coworkers. This difficulty, which was also discussed in Ref. [14], provides the motivation for our present work. In this regard, our scheme now provides definitive, quantitative access to the full 3D nanophotonic environment under general conditions.

We don't want to oversell the point that there are problems with the Nicoletti approach (the results published in their paper are nevertheless beautiful), yet we think that there is a consensus in the scientific community that their approach is problematic in many respects.

Without these points clarified by the authors, I cannot recommend the publication of the manuscript. In my opinion, the current work uses modeling so heavily that there is little to none experimental data/information in the final 3D plasmonic maps.

Our response:

It appears that the Reviewer has somewhat misinterpreted the approach taken in our work. We agree that a pure fitting procedure, as outlined by the Referee, would not constitute an overly appealing approach. However, we hope that our above discussion can convince the Reviewer that we are doing something significantly more sophisticated here, namely: we (i) provide a generic but well-chosen basis for the plasmonic fields, and (ii) determine the expansion coefficients using a compressed sensing optimization together with the tilted experimental EELS maps. We have developed this scheme over the last years and have now refined it to the stage where it can be employed to experimental data. Maybe we have not been careful enough in the description of this approach.

Actions taken:

In view of the difficulties of the Reviewer, we have now significantly expanded our discussion about the rationale behind our scheme. We hope that with these modifications the criticism raised in the report is refuted, and the Reviewer can now support publication.

Reviewers' comments:

Reviewer #2 (Remarks to the Author):

Comments of the manuscript: NCOMMS-16-24742

Dear authors, thanks for your replies to my queries. Most of them have been solved.

I would like to point out that the explanation of the origin of the shift of the plasmon peaks is still weak. Your hypothesis may be correct but they are based on certain qualitative assumptions. Nevertheless, I think that it does not detract the full scheme described in the paper.

1. Reviewer: Moreover, the authors also mention that although they observed energy shifts, "the modal distribution in the plasmon resonance map did not change significantly". This is an interesting observation because retardation effects and not contamination may be the cause of the red-shift. In general retardation fields affects the energy and the amplitude of a resonant mode but not its projected symmetry. At high tilts, like in the experiments performed here the corresponding resonant fields have more coupling to the Si₃N₄ substrate and could red-shift. Directionality in plasmonics is important [Coenen et al. Nano Lett 2011, 11]. Also the substrate should break z symmetry, and mismatches between reprojections and experimental maps may be larger as trajectories become more parallel to the substrate [Nicoletti et al. Nature 2013].

Response of authors:

This is an interesting idea and it would be great to be able to measure these direction-induced shifts.

However, this requires pristine, unaltered samples to be able to exclude contamination effects. We cannot exclude directionality effects in our measurements, yet we are quite sure that contamination has a much larger impact due to the following reasons:

-We compared peak position at 0° tilt angle before, during and after the experiment and observed the peak shift (Fig. S6).(Fig S7?) To clarify this, we have now indicated the peak energy in Fig. S6.

-The shift of the peak positions during acquisition (starting at a large negative angle to 0° to a large positive angle) appears quite linear (Fig. S5).(Fig s6?)

This observation may be the missing clue that I misunderstood in the first revision. If the series of images was acquired from "a large negative angle to 0° to a large positive angle" and if the peak shift did not change sign (from red to blue-shift for example) then the origin of the effect may well be the layer of carbon contamination. Then, to make clear this, I suggest that you label Figure s7 showing the tilt angles that correspond to each spectra. This may help to the next reader of your paper to avoid the confusion I had.

It would be great if you could also add some hypothesis that explain the role of a carbon layer of contamination:

- How the carbon layer can shift surface plasmons? Any hypothesis? Any physical origin that would explain it?
- Do the authors know about any previous reference in the literature of EELS in which a red shift in plasmon peaks is observed due to carbon contamination?
- Is an effect of the volume plasmon of the carbon layer of contamination (that will grow with

time) which adds a background signal that in turn will distort the much smaller surface plasmon peaks?

Reviewer #3 (Remarks to the Author):

Dear Editor,

Most of my questions in the previous review were answered in the authors' rebuttal document. However, I still think that the use of "tomographic reconstruction" to describe the current technique is not 100% percent accurate, as the final 3D results are mixture of theory and experiment (as authors also agree in the rebuttal document). This description (which is in the title of the manuscript) may cause certain level of confusion for readers, as the established wisdom in the field is that "tomographic reconstruction" represents purely experimental 3D results. However, this is not a strong enough reason for rejection. With this warning expressed, the manuscript can be published in its current form.

Reviewer #2

We thank the referee for the careful reading of the manuscript and for suggesting further changes to improve the readability of our manuscript. We have followed his/her suggestion and have added the tilt angles in Supplementary Fig. 6 (we also thank the referee for pointing out the misprints regarding the supplementary figures).

As regarding the other points of the referee:

- An increase of the embedding dielectric constant typically leads to a decrease of the plasmon energy, see e.g. Becker et al., *Plasmonics* 5, 161 (2010). The same decrease is also expected for a thin carbon layer (with a permittivity around 3 in the visible regime), as can be varified from simulations, see e.g. Habberfehlner et al., *Nano Lett.* 15, 7726 (2015).
- A similar red-shift due to carbon contamination was observed by Nicoletti et al. (see Methods > Spectral processing, as well as main text).
- In Nicoletti et al., Fig. 1b, the authors estimate the effect of carbon contamination on the plasmonic spectra. As can be seen there, contamination results in a broad background that should have little impact on our tomography scheme.

Actions taken

We added a short discussion about contamination-induced red-shifts in the Supplementary Information, along with references to literature. The tilt angles have been added in Supplementary Fig. 6.

Reviewer #3

We thank the referee for now recommending publication in Nature Communications. We have followed his/her suggestion and have changed the title to "Tomographic imaging of the photonic environment of plasmonic nanoparticles".

We note that in the specialized literature the phrase "tomography" is commonly used in combination with the solution of inverse problems. For instance, the manuscript "Global seismic tomography: the inverse problem and beyond" by J. Trampert, *Inverse Problems* 14, 371 (1998) discusses tomography through solution of an inverse problem in seismology. Also in the book "Waves and Fields in Inhomogeneous Media by W. C. Chew, IEEE Press on Electromagnetic Waves (New York, 1995), in Chapter 9 the author starts with "In inverse scattering, one attempts to infer the properties of the scatterer from the scattered fields ..." and then clearly states "The science of reconstructing an object from some measurement data is known as tomography". It is our understanding that the criticism of the reviewer concerns the phrase "tomographic reconstruction" which has now been changed to "tomographic imaging".

Actions taken

We have changed the title to "Tomographic imaging of the photonic environment of plasmonic nanoparticles".

REVIEWERS' COMMENTS:

Reviewer #2 (Remarks to the Author):

Dear authors, thanks for the answers.
From my side, I have nothing else to comment.